# Fine Mapping of *qTGW7b*, a Minor Effect QTL for Grain Weight in Rice (*Oryza sativa* L.)

**DOI:** 10.3390/ijms23158296

**Published:** 2022-07-27

**Authors:** Houwen Gu, Kunming Zhang, Sadia Gull, Chuyan Chen, Jinhui Ran, Bingyin Zou, Ping Wang, Guohua Liang

**Affiliations:** 1Jiangsu Key Laboratory of Crop Genetics and Physiology/Key Laboratory of Plant Functional Genomics of the Ministry of Education, Agricultural College, Yangzhou University, Yangzhou 225009, China; houwengu@126.com (H.G.); 18247161873@163.com (K.Z.); sadiagull2022@163.com (S.G.); chuyanchen007@126.com (C.C.); jinhuiran2022@126.com (J.R.); 18906365353@163.com (B.Z.); m15252573991@163.com (P.W.); 2Jiangsu Co-Innovation Center for Modern Production Technology of Grain Crops, Agricultural College, Yangzhou University, Yangzhou 225009, China; 3Joint International Research Laboratory of Agriculture and Agri-Product Safety, Institutes of Agricultural Science and Technology Department, Yangzhou University, Yangzhou 225009, China

**Keywords:** rice (*Oryza sativa* L.), grain weight, minor effect QTL, *qTGW7b*, fine mapping

## Abstract

Grain weight is a key trait that determines rice quality and yield, and it is primarily controlled by quantitative trait loci (QTL). Recently, attention has been paid to minor QTLs. A minor effect QTL *qTGW7* that controls grain weight was previously identified in a set of chromosomal fragment substitution lines (CSSLs) derived from Nipponbare (NPB)/93-11. Compared to NPB, the single segment substitution line (SSSL) N83 carrying the *qTGW7* introgression exhibited an increase in grain length and width and a 4.5% increase in grain weight. Meanwhile, N83 was backcrossed to NPB to create a separating population, *qTGW7b*, a QTL distinct from *qTGW7*, which was detected between markers G31 and G32. Twelve near-isogenic lines (NILs) from the BC_9_F_3_ population and progeny of five NILs from the BC_9_F_3:4_ population were genotyped and phenotyped, resulting in the fine mapping of the minor effect QTL *qTGW7b* to the approximately 86.2-kb region between markers G72 and G32. Further sequence comparisons and expression analysis confirmed that five genes, including *Os07g39370*, *Os07g39430*, *Os07g39440*, *Os07g39450*, and *Os07g39480*, were considered as the candidate genes underlying *qTGW7b*. These results provide a crucial foundation for further cloning of *qTGW7b* and molecular breeding design in rice.

## 1. Introduction

Rice represents the primary food consumed by urban and rural residents in China, accounting for more than 60% [1]. Rice production is a crucial strategy for combating climate change and population growth. Rice yield is primarily determined by the following three factors: panicle number per plant, number of grains per panicle, and grain weight. Rice grain weight is affected by grain length (GL), grain width (GW), grain thickness (GT), grain filling, and other factors [2]. Multiple genes synergistically regulate the quantitative trait of grain weight to maintain the dynamic balance of rice yield. Many quantitative trait loci (QTLs) for rice grain weight have been mapped to chromosomes, as a result of the enrichment of molecular markers by whole genome sequencing and the development of novel mapping methods [3,4,5,6,7].

With the advancement of functional genomics and molecular genetics in rice, many genes/QTLs that control grain size have been cloned and functionally studied, including *GS3*, encoding the G-protein γ subunit, which reduces grain length by competitively interacting with Gβ [8]. DEP1/qPE9-1 also encodes a γ subunit of a G-protein and GS3 and DEP1 can interact with OsMADS1 and function as cofactors in the regulation of common target genes that control grain size [8,9]. *GW6* encodes a GAST (GA-stimulated) family protein that regulates gibberellin response and biosynthesis to affect grain width and weight positively [10]. *GL3.1/qGL3*, encoding a serine/threonine phosphatase, is a negative regulator of grain length that changes cell numbers by directly phosphorylating the cell cycle protein cyclin-T1;3 [11,12]. *TGW6*, which encodes an IAA-glucose hydrolase, controls the transition of IAA from the syncytial to the cytosolic stage to negatively regulate seed length [13]. *LG1* encodes a ubiquitin-specific protease, OsUBP15. OsDA1 interacts with OsUBP15 to reduce its stability, negatively regulating cell proliferation and rice seed size [14]. *GS2/OsGRF4* encodes a growth-regulating factor that suppresses cell elongation and cytokinesis by interacting with transcriptional co-activators OsGRFs, thereby influencing grain size and weight. In addition, OsmiR396 regulates *GS2*, and *GS2^AA^* mutations can affect OsmiR396-mediated splicing, resulting in the large-grain phenotype [15,16]. *GW5* encodes a calmodulin-binding protein and negatively regulates grain width and grain weight. *GW5* inhibits the autophosphorylation of GSK2 and the phosphorylation of OsBZR1 and DLT by GSK2, affecting the accumulation of unphosphorylated OsBZR1 and DLT proteins in the nucleus [17]. *TGW3/qTGW3/GL3.3*, encoding a GSK3/SHAGGY-like family kinase, OsSK41/OsGSK5, interacts with and phosphorylates OsARF4 to regulate rice grain size and weight via auxin-responsive signaling. Loss of function of *qTGW3* results in larger rice grains [18,19,20]. These genes act as negative regulators on rice grain size.

In contrast, some genes control grain length as positive regulators. *GS5* encodes a serine carboxypeptidase, a quantitative trait gene that controls grain width, fullness and thousand-grain weight in rice. A high level of *GS5* expression results in larger seeds [21]. *GW2* on chromosome 2 encodes a cyclic E3 ubiquitin ligase that inhibits cell division by anchoring its substrate to the proteasome for degradation to regulate seed size [22,23]. *GL7/GW7*, which consists of two tandem repeats, *GL7-S1* and *GL7-S2*, regulates grain length by increasing cell division in the longitudinal direction and controls grain width by decreasing cell division in the transverse direction. GL7 interacts with the cell proliferation-promoting regulator OsSPL16/GW8. In rice, a higher level of *OsSPL16* expression promotes cell division and grain filling, increasing grain width and yield [24,25]. *qLGY3*, located on chromosome 3 and encoding the MADS-domain transcription factor OsMADS1, regulates grain size by interacting with GS3 and DEP1 to enhance *OsMADS1* transcriptional activity [9]. *WG7*, which encodes a CW domain-containing zinc finger protein, directly binds to the promoter of *OsMADS1* and promotes *OsMADS1* expression by enhancing its H3K4me3 enrichment to positively regulate grain size in rice [26].

Nonetheless, most QTLs that control grain size and weight show minor effects and are difficult to clone. To date, only six QTLs with minor effects have been cloned. Five control rice flowering time and one controls rice grain weight. The additive effects of the five QTL related to heading time (*DHD4*, *qHd1*, *Hd17*, *Hd18*, and *DTH2*) ranged from 1 to 4 days [27,28,29,30,31]. In different years, the additive effect of the grain weight QTL *qTGW1.2b* ranged from 0.13 g to 0.19 g [32], which was significantly less than the additive effect of the cloned major effect QTL on grain weight. Minor effect QTL are sensitive to genetic background and strongly influenced by environmental and measurement biases. Consequently, cloning minor effect QTL has proven to be a formidable challenge.

In our previous studies, one minor effect QTL for grain weight *qTGW7* was detected on the long arm of chromosome 7, using a set of CSSLs derived from a cross between *indica* rice cultivars 93-11 and a *japonica* cultivar NPB [33]. In this study, we fine mapped *qTGW7b*, a QTL with a minor effect regulating grain weight, close to *qTGW7* in an 86.2-kb region. Our research expands the minor effect QTL for grain size regulation in rice and lays the groundwork for future gene cloning and functional studies.

## 2. Results

### 2.1. Grain Size Analysis of NPB and N83

The detection of a minor effect QTL *qTGW7* controlling TGW has been reported previously using a set of CSSLs constructed with NPB as the recipient parent and 93-11 as the donor parent [33]. N83, a line containing a 93-11 fragment, has a larger grain size than NPB. The substitution segments of N83 that originate in 93-11 were detected by resequencing, and only a substitution segment from 15.9 M to 23.8 M on the chromosome 7 long arm was identified (Figure 1). The GL and GW of N83 increased by 4.3% and 5.5% in 2020 and 4.3% and 5.6% in 2021, respectively, compared to the recurrent parent NPB (Figure 2A–C). However, there was no difference in GT between NPB and N83 (Figure 2D). Furthermore, the dehulled grain of N83 was significantly bigger than that of NPB (Figure 2A), resulting in a 4.4% and 4.6% increase in the TGW of N83 relative to NPB in 2020 and 2021, respectively (Figure 2E, Appendix A). We also investigated the mature seed size of 93-11 and NPB. In addition, 9311 had longer GL, narrower GW and GT, and higher TGW than NPB (Appendix A). These results indicate that the differences in grain size and weight between N83 and NPB are stable and that there may be a QTL for N83 that influences grain size and weight. 

### 2.2. Cytological Analysis of Spikelet Hull in NPB and N83

The spikelet hull (consisting of palea and lemma) imposes a maximum size restriction on the final grain [22,34]. To investigate the factors responsible for the increase in GL and GW of the substitution line N83, scanning electron microscopy (SEM) was used to evaluate the cell morphology of the lemma by comparing the cell length and width of the outer epidermal cells in spikelet hulls in NPB and N83. As demonstrated in Figure 3, the outer epidermal cells of N83 spikelet hulls were longer and wider than those of NPB spikelet hulls. Counting the number of cells in the spikelet hull in the direction of grain length revealed no difference between N83 and NPB (Figure 3). These results indicate that the increased cell expansion in spikelet hulls is primarily attributable to the N83 phenotype.

### 2.3. Genetic Analysis of Grain Size-Related Traits of N83

To validate *qTGW7* and determine whether this QTL influenced the variation in N83 grain size, we backcrossed N83 with NPB and then self-crossed to create a BC_9_F_2_ segregating population. The BC_9_F_2_ segregation population contained 289 individual plants for linkage analysis and QTL analysis. One heterozygous plant was selected, covering G6-G32 for self-crossing to generate the BC_9_F_3_ population. The BC_9_F_3_ population consisted of 297 individual plants. The Anderson–Darling test, D’Agostino and Pearson test, Shapiro–Wilk test, and Kolmogorov–Smirnov test were utilized to analyze the distribution of the grain size-related traits GL, GW, and TGW. Remarkably, phenotypic variation in GL and TGW approximated a normal distribution based on quantile-quantile plot (QQ plot) values (Figure 4 and Appendix A). Transgressive segregation was observed only for GW (Table 1 and Appendix A).

Combining the genotype and phenotype information of each plant in the BC_9_F_3_ population, frequency distributions for each genotype group were plotted as a series, and differences were observed between the GL, GW, and TGW genotype groups. The 93-11 homozygotes were clustered towards the area of higher values, while the NPB homozygotes to the area of lower values (Figure 5, Table 2), indicating that there are QTL affecting grain size in this population and it is suitable for QTL analysis. In addition, the correlation analysis revealed a relationship between TGW, GL, and GW. All three correlation coefficients were highly significant, including 0.647 between GL and GW, 0.719 between GL and TGW, and 0.709 between GW and TGW (Table 3).

### 2.4. Validation and Fine Mapping of qTGW7b

TGW was selected as the genotype for QTL analysis, since the differences between N83 and NPB in GL and GW were minor, susceptible to environmental influences, and difficult to discern in the field. In contrast, the variation in TGW in the segregated population was as high as 5 g. Therefore, two linkage maps were constructed, combining each plant’s genotype and phenotype information in the BC_9_F_2_ and BC_9_F_3_ populations (Table 4). Significant QTL effects for the TGW were observed in both BC_9_F_2_ and BC_9_F_3_ populations. For TGW, the enhancing alleles were derived from 93-11 in the two populations. In BC_9_F_2_, the additive effects were 0.69 g for TGW, explaining 19.33% of the phenotypic variances. In BC_9_F_3_, the additive effects were 0.61 g for TGW, explaining 17.17% of the phenotypic variations. The results suggested there might be a QTL in the segregating region.

In addition, twelve different types of recombinants were screened from the BC_9_F_3_ population based on the positions of recombination breakpoints determined using molecular markers, and individual recombinant BC_9_F_4_ plants with homozygous 93-11 genotypes were selected and developed into NILs (R1 to R12) in order to quantify the TGW phenotype. The TGW of plants in groups R1 through R7 was not significantly different from NPB. Similarly, there were no significant differences between plants in groups R8 through R12 and N83. In contrast, the TGW of plants that belonged to groups R1 to R7 and R8 to R12 varied considerably (Table 5). By combining the genotypes and phenotypes of twelve NILs, we were able to map *qTGW7* to a 651-kb genomic region flanked by markers G31 and G32 (Figure 6A). However, this interval differs from the previously reported one, so we designated it as *qTGW7b*.

Ten BC_9_F_3_ heterozygous plants were utilized to establish a self-population for fine mapping in order to narrow down the mapped region further. We identified ten recombinants by screening 3118 BC_9_F_3:4_ generation individuals for the G31 and G32 markers (Figure 6B). Using five additional polymorphism markers, the ten recombinants were separated into five groups based on the recombination breakpoints (Figure 6B). A test of the progeny of recombinant NPB homozygotes and 93-11 homozygotes from the BC_9_F_3:4_ generation revealed a significant difference in TGW between the N2 and N5 groups; however, this was not among the N1, N3, or N4 groups (Table 6). *qTGW7b* was ultimately narrowed to a ~86.2-kb region flanked by markers G72 and G32 (Figure 6B).

### 2.5. Analysis of qTGW7b Candidate Genes

According to the Rice Genome Annotation Project (http://rice.uga.edu/, accessed on 20 January 2022), the 86.2 kb region contains 14 predicted genes (Table 7). These genes encode four retrotransposon proteins (*Os07g39380*, *Os07g39390*, *Os07g39410*, and *Os07g39420*), three transporter family proteins (*Os07g39350*, *Os07g39360*, and *Os07g39460*), two expressed proteins (*Os07g39370* and *Os07g39450*), a polyol transporter protein (*Os07g39400*), an mTERF family protein (Os07g39430), a zinc finger protein (*Os07g39440*), a chitin-inducible gibberellin-responsive protein (*Os07g39470*), and a WRKY transcription factor (*Os07g39480*). First, the retrotransposon proteins were eliminated as candidates for *qTGW7b*. Genomic sequencing of the last ten candidate genes in NPB and N83 revealed that amino acid diversity occurred, except for *Os07g39350*, *Os07g39360*, *Os07g39460*, and *Os07g39470*. Therefore, the remaining six candidate genes (*Os07g39370*, *Os07g39400*, *Os07g39430*, *Os07g39440*, *Os07g39450*, and *Os07g39480*) were prioritized for further investigation. Os07g39370 had a 4-bp insertion, a 17-bp deletion and a single SNP, which resulted in a code shift beginning at the 47th amino acid. *Os07g39400* contained a 12-bp insertion and four SNPs, which caused four amino acid insertions and one nonsynonymous mutation. Three SNPs were present on *Os07g39430*, resulting in one synonymous and two nonsynonymous mutations. Four SNPs were found in both *Os07g39440* and *Os07g39450*, resulting in one synonymous and three nonsynonymous mutations. *Os07g39480* contained a 42-bp deletion, a 4-bp deletion and three SNPs, which led to four nonsynonymous mutations and the deletion of amino acids at positions 479 to 492 and 543 (Table 7).

We further investigated the expression of these six genes in developing young panicles and found that only *Os07g39400* exhibited a non-significant difference in expression between NPB and N83. *Os07g39480* had the highest expression level among the remaining differentially expressed genes. *Os07g39370* and *Os07g39480* had significantly lower expression levels in N83 than in NPB, whereas *Os07g39430*, *Os07g39440*, and *Os07g39450* had significantly higher expression levels in N83 than in NPB (Figure 7). These results indicate that five genes, including *Os07g39370*, *Os07g39430*, *Os07g39440*, *Os07g39450*, and *Os07g39480,* were considered the candidate genes underlying *qTGW7b*.

## 3. Discussion

Since the beginning of this century, great progress has been made in cloning major QTLs for grain size-related traits in rice, allowing for the possibility of novel breeding strategies based on genotype information [35,36]. Nevertheless, the QTLs that have been cloned and fine mapped represent only a small proportion of the QTLs detected in the primary populations [37]. It has been assumed that the primary-mapped QTLs are far below the actual basis of trait variation, as most experiments have a low capacity for detecting minor effect QTLs [38]. In this study, we fine mapped a new minor effect QTL *qTGW7b* that controls grain weight in an 86.2-kb region in chromosome 7. 

According to the report, our group has previously identified a minor effect QTL *qTGW7* that controls rice grain weight on chromosome 7 between 19.5 M and 22.5 M using the same set of CSSLs [33]. In contrast, the fine mapping interval of *qTGW7b* in this study deviates from the primary mapping interval, which may be attributable to a number of factors. First, minor effect QTLs have minor effects and can be misidentified phenotypically. Additionally, marker coverage can influence the precision of QTL analysis. Any single contradictory plant during the minor effect QTL detection procedure can result in bias in the QTL mapping region. The above concerns could be why the *qTGW7* detection region is 0.9 M to the left of *qTGW7b* on the chromosome. Additionally, *qGL7-2,* a major effect QTL for grain length, was also detected in the region adjacent to *qTGW7b* [39]. Minor effect QTL are often difficult to detect in such a background.

CSSL is a series of continuous target interval hybrid background homozygous NIL obtained by hybridization, self-crossing, backcrossing, and marker-assisted selection (MAS). Using CSSL can eliminate the interference of background and environment on QTL mapping and enable multi-point repeated use over many years, which is widely utilized in QTL mapping [40,41]. In this study, N83 was a single-segment substitution line (Figure 1). The individual plants in the BC_9_F_2_ or BC_9_F_3_ segregating population of the N83/NPB backcross differed only within the introgression segment of chromosome 7, without background differences. Thereby, phenotypic variations were only caused by QTL within this segment.

The development of functional genomics and molecular genetics in rice will permit the functional validation of candidate genes and molecular mechanistic studies. Consequently, the cloning of QTL for critical agronomic traits in rice is highly dependent on the precision of fine mapping. Our research offers a novel strategy that only requires the detection of a small number of recombinants based on DNA markers in the region of interest. In the beginning, twelve BC_9_F_3_ plants with distinct breakpoints in the 8-Mb target region were initially identified. One QTL, designated *qTGW7b*, was identified within a 651-kb interval using segregating populations derived from self-crossing progeny (Figure 6A). Afterwards, five BC_9_F_3:4_ plants with distinct breakpoints in the putative *qTGW7b* region were chosen and self-crossed to generate populations that segregated. As significant TGW differences between NPB and 93-11 homozygotes were detected in N2 and N3 that were segregated in different regions, the *qTGW7b* was narrowed to an 86.2-kb interval (Figure 6B).

For *qTGW7b* that has been delimitated within an 86.2-kb interval, the QTL region contained 14 annotated genes. Sequencing and expression analysis identified the candidate genes for *qTGW7b* as *Os07g39370*, *Os07g39480*, *Os07g39430*, *Os07g39440*, and *Os07g39450*. Both *Os07g39370* and *Os07g39450* encode unidentified proteins. *Os07g39430* encodes a mitochondrial transcription termination factor, regulating the transcription and translation of genes. *Os07g39440* encodes a zinc finger protein. Several studies indicate that zinc finger proteins are also involved in regulating the size and weight of rice grains. *Drought and salt tolerance* (*DST*), which encodes a zinc-finger transcription factor, regulates the expression of *Gn1a/OsCKX2* in the reproductive meristem (SAM) directly. *DST^reg1^*, a semi-dominant allele of the *DST* gene, interferes with DST-directed regulation of *OsCKX2* expression and raises CK levels in the SAM, resulting in increased panicle branching, grain number, and grain weight [42]. *Lacking rudimentary glume 1* (*LRG1*) encodes a zinc-finger protein that plays a crucial role in the regulation of spikelet organ identity and grain size. *LRG1* mutation results in smaller grain size and diminished grain weight [43]. *Os07g39480* encodes a WRKY transcription factor, OsWRKY78, which is involved in the regulation of grain size in rice, and *OsWRKY78*-RNAi has a smaller grain size and decreased grain weight [44].

According to quantitative genetic theory and modern QTL mapping progress, minor effect QTL play an important role in the regulation of important agronomic traits in rice [45], and this QTL should not be neglected in mechanistic analysis or breeding applications. In this study, the *qTGW7b* allele from 93-11 significantly increased GL and GW compared to that of *qTGW7b^NPB^*, resulting in a 4.5% increase in TGW (Figure 2, Table 1). The morphology of brown rice became larger without any changes in the rice appearance quality (Figure 2A). Additionally, plants with the *qTGW7b^93−11^* had more GN and longer GL (Appendix A, Appendix A). Furthermore, *qTGW7b^93−11^* plants have thicker and stronger stems (Appendix A), which contribute to their resistance to lodging. These are all beneficial traits for yield composition. Therefore, *qTGW7b* can be a valuable genetic resource for grain and plant improvement in rice.

## 4. Materials and Methods

### 4.1. Plant Materials and Growth Conditions

A set of CSSLs was generated by using NPB, a *japonica* cultivar, as the recipient and 93-11, an *indica* cultivar, as the donor, as previously reported [46]. N83, a BC_8_F_6_ line, was backcrossed with NPB to generate a segregating BC_9_F_2_ population for grain weight QTL analysis and fine mapping. Twelve sets of NILs originated from the BC_9_F_3_ population, and five NILs populations were derived from a large BC_9_F_3:4_ population developed for fine mapping (Figure 8). All plants were grown in paddy fields at Huaisi county of Yangzhou (Jiangsu province, China) or Lingshui (Hainan province, China) under normal cultivation conditions.

### 4.2. DNA Extraction, PCR, and Marker Development

Total DNA was extracted from young rice leaves of each plant with the cetyltrimethylammonium bromide (CTAB) method [47]. The polymerase chain reaction (PCR) was conducted in a 20 µL volume with 10 µL of 2 × Taq Master Mix (Vzayme, Nanjing, China), 2 µL of DNA template, and 6 µL of ddH_2_O. The PCR products were separated by electrophoresis on a 3% agarose gel. All insertion/deletion (Indel) markers used for QTL mapping were developed in accordance with the RiceVarmap 2 website (http://ricevarmap.ncpgr.cn/, accessed on 15 March 2021) by comparing the sequences in the target region between NPB and 93-11 and were detected using agarose gel electrophoresis. All of the Indel markers are listed in Appendix A.

### 4.3. Scanning Electron Microscope (SEM)

Young spikelets about to be extracted from flag leaf sheaths were collected, fixed with FAA, and vacuumized in a vacuum dryer for 30 min. After dehydration in a gradient of ethanol (30% alcohol, 60% alcohol, 75% alcohol, 85% alcohol, 90% alcohol, 96% alcohol, and 100% ethanol) once for 30 min, three times in pure ethanol, the sample was transferred to the sample box with a critical point dryer for drying treatment. The treated samples were fixed on the pedestal and sprayed with gold, and then the three different parts of the grain were observed by scanning electron microscopy.

### 4.4. Sequence Analysis

The RGAP (Rice Genome Annotation Project, http://rice.uga.edu/ accessed on 20 January 2022) database was used to annotate the 14 open reading frames (ORFs) and their putative functions in the 86.2-kb region between G72 and G32. The sequencing primers for the ORFs were designed based on the NPB sequences in the Phytozome database (https://phytozome-next.jgi.doe.gov/, accessed on 20 January 2022). Phanta Max Master Mix was utilized to formulate the PCR reaction mixture (Vazyme, Nanjing, China). DNASTAR software was used to analyze the sequencing results. Appendix A of the supplementary materials details the amplifying primers for candidate genes.

### 4.5. Phenotype Investigation

The agronomic traits included plant height (PH), panicle number per plant (PN), panicle length (PL), grain number per panicle (GN), grain length (GL), grain width (GW), grain thickness (GT), thousand-grain weight (TGW), and grain yield per plant (GYP). PH represents the height from the ground to the tip of the highest panicle. PL was measured from panicle neck to panicle tip. Ten fully filled grains were selected for investigation of GL, GW, GT, and TGW. Twenty individual plants of NPB and N83 were chosen randomly for PH, PN, PL, GN, GL, GW, GT, TGW, and GYP. Linkage and QTL analysis were performed on TGW of BC_9_F_2_ and BC_9_F_3_ segregation populations.

### 4.6. Quantitative Real-Time PCR (qRT-PCR)

For the candidate gene analysis in the target region, young panicles from NPB and N83 were collected. Three plants were harvested. After total RNA was extracted using an RNA extraction kit (Tiangen, Beijing, China), 20 µL of cDNA was synthesized by a one-step RT-qPCR kit (Tiangen, Beijing, China). qRT-PCR was performed using ChamQ Universal SYBR qRT Master Mix (Vazyme, Nanjing, China) in 20 µL. All qRT-PCR primers are listed in Appendix A.

### 4.7. Data Analysis

The phenotypic differences between NPB and N83 and various NILs were evaluated with Student’s *t*-tests. SPSS Statistics (Versions 25.0, IBM Corporation, Armonk, New York, NY, USA) software was used to analyze the correlation between GL, GW, and TGW. All agronomic traits were investigated at maturity. Paddy grains were dried naturally after harvest for at least one week before testing.

## 5. Conclusions

As an essential trait of yield, grain size has a direct effect on the production of rice. There are few reports on the mapping of grain size QTL with a minor effect. In this study, a minor effect QTL *qTGW7b* that controls grain weight was finely mapped in an 86.2-kb region between markers G72 and G32 on chromosome 7 by constructing a backcross segregating population and NILs, using a set of CSSLs produced by NPB/93-11. Cytological analysis revealed that the *qTGW7b* allele from 9311 increased cell proliferation in the spikelet, resulting in a 4.5% increase in grain weight. *qTGW7b^93−11^* also increased stem thickness, GN, GL, and yield. Fine mapping of *qTGW7b* laid the foundations for further gene cloning and provided a new genetic resource for grain and plant architecture improvement.

## Figures and Tables

**Figure 1 ijms-23-08296-f001:**
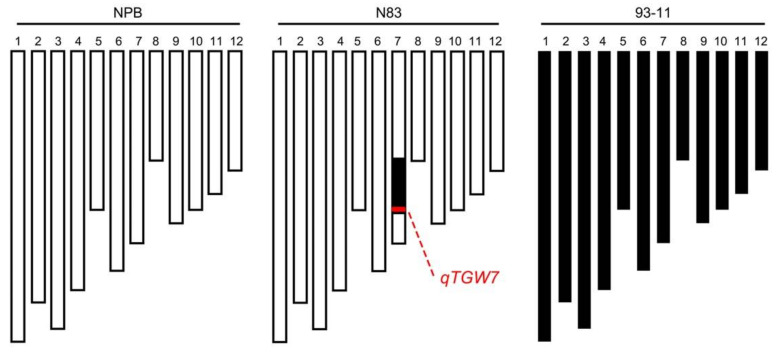
Graphical genotypes of NPB, N83, and 93-11. Twelve chromosomes are represented by vertical bars and numbered at the top. In the N83 chromosomal maps, the black regions indicate the 93-11 segments in the NPB background, and the red region indicates the region containing the *qTGW7* locus, as previously reported.

**Figure 2 ijms-23-08296-f002:**
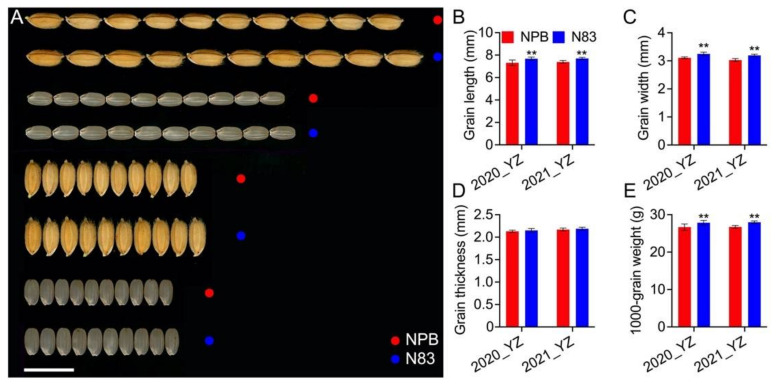
Grain characterization of NPB and N83. (**A**) Mature and dehulled seeds of NPB and N83. Scale bars, 1 cm. Comparison of GL (**B**), GW (**C**), GT (**D**), and TGW (**E**) in two years. Data in (**A**–**E**) (*n* = 20) are means ± SD, ** *p* < 0.01. Student’s *t*-test was used to generate the *p* values.

**Figure 3 ijms-23-08296-f003:**
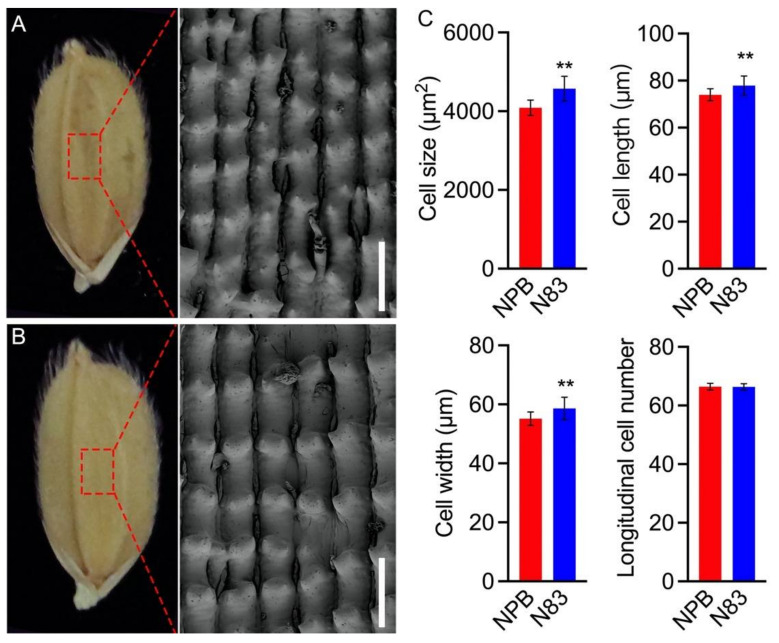
Images of spikelet outer epidermal cells were obtained by SEM in NPB (**A**) and N83 (**B**). Scale bars, 100 μm. (**C**) Comparisons of cell size, length, width, and longitudinal cell number of the spikelet hulls in NPB and N83. Data in (**A**) (*n* = 30) are means ± SD, ** *p* < 0.01. Student’s *t*-test was used to generate the *p* values.

**Figure 4 ijms-23-08296-f004:**
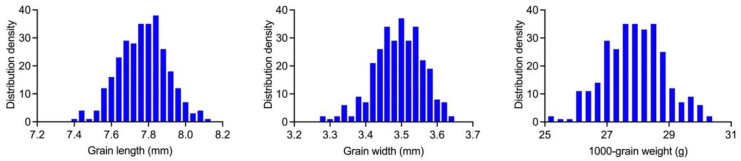
Frequency distributions of GL, GW, and TGW in the BC_9_F_3_ population.

**Figure 5 ijms-23-08296-f005:**
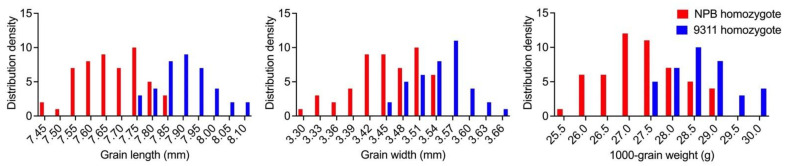
Frequency distribution of GL, GW, and TGW of NPB and 9311 homozygotes in the BC_9_F_3_ population.

**Figure 6 ijms-23-08296-f006:**
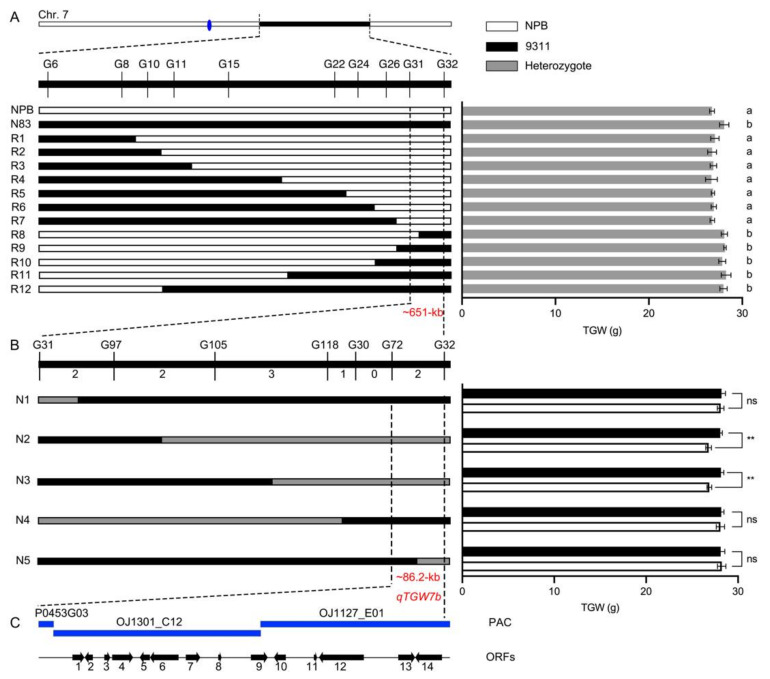
QTLs for grain weight that were found to be segregated in the rice populations. (**A**) Primary mapping of *qTGW7b*. The blue oval on the chromosome indicates a centromere. Statistical differences are labeled with “a” and “b” using LSD test (*p* < 0.05, one-way ANOVA). (**B**) Fine mapping of *qTGW7b*. Student’s *t*-test was used to generate the *p* values. Data are means ± SD, ns, *p* > 0.05, ** *p* < 0.01. (**C**) Candidate genes in the *qTGW7b* region. Blue rectangles represent PAC clone.

**Figure 7 ijms-23-08296-f007:**
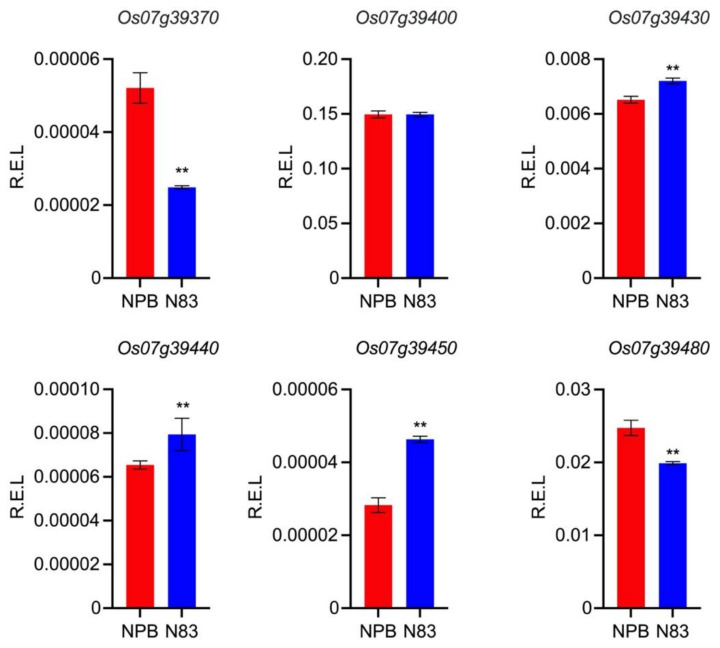
Expression analysis of *qTGW7b* candidate genes. Data are mean ± SD with three technical replicates, Student’s *t*-test was used to generate the p values, ** *p* < 0.01. R.E.L: Relative expression level.

**Figure 8 ijms-23-08296-f008:**
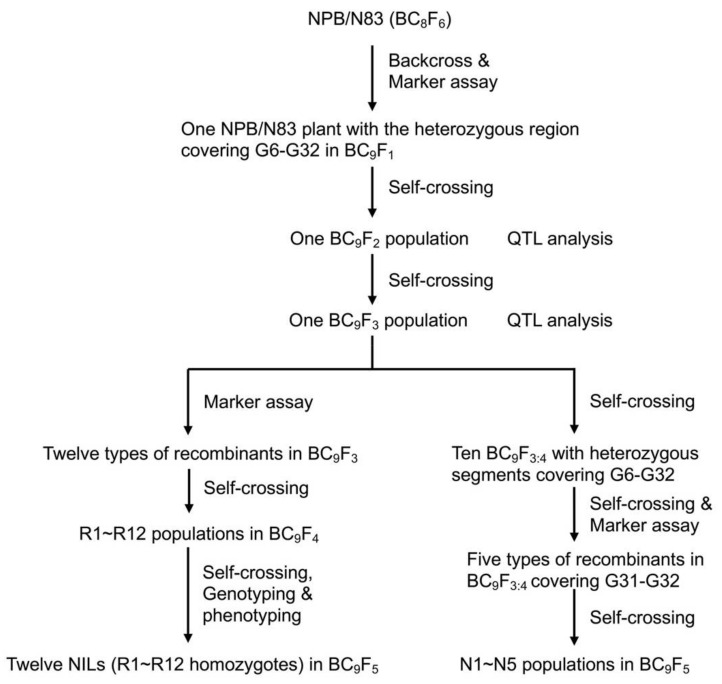
Development of the rice populations used in this study.

**Table 1 ijms-23-08296-t001:** Phenotypic performance of GL, GW, and TGW in the BC_9_F_3_ population.

Trait	Range	Mean	SD	Skewness	Kurtosis
GL	7.40–8.14	7.78	0.13	−0.06	−0.09
GW	3.28–3.65	3.50	0.07	−0.42	0.22
TGW	25.30–30.30	27.9	0.97	0.02	−0.18

GL, grain length (mm); GW, grain width (mm); TGW: 1000-grain weight (g).

**Table 2 ijms-23-08296-t002:** Phenotypic performance of GL, GW, and TGW in NPB and 93-11 homozygotes.

Trait	Haplotype	Range	Mean	SD	Skewness	Kurtosis	*p* Value
GL	NPB	7.44–7.85	7.67	0.1	−0.21	−0.63	<0.0001
	93-11	7.73–8.09	7.90	0.09	0.24	−0.17	
GW	NPB	3.30–3.55	3.45	0.06	−0.48	−0.27	<0.0001
	93-11	3.44–3.65	3.54	0.05	−0.12	−0.10	
TGW	NPB	25.31–29.11	27.3	0.90	0.11	−0.48	<0.0001
	93-11	27.45–30.04	28.59	0.72	0.39	−0.65	

TGW, 1000-grain weight (g); GL, grain length (mm); GW, grain width (mm). Student’s *t*-test was used to generate the *p* values.

**Table 3 ijms-23-08296-t003:** Correlation analysis of GL, GW, and TGW in the BC_9_F_3_ population.

	GL		GW		TGW	
	r	Pr	r	Pr	r	Pr
**GL**	1.000		0.647	***	0.719	***
**GW**	0.647	***	1.000		0.709	***
**TGW**	0.719	***	0.709	***	1.000	

Pr: Pearson’s correlation coefficient. *t* test was used to generate the *p* value. *** *p* < 0.001.

**Table 4 ijms-23-08296-t004:** QTL detected for TGW in the BC_9_F_2_ and BC_9_F_3_ population.

Population	Number of Plants	Segregating Region	LOD	*A*	*D*	R^2^ (%)
BC_9_F_2_	289	G31-G32	13.81	0.69	0.15	19.33
BC_9_F_3_	297	G31-G32	12.56	0.61	0.12	17.17

A, additive effect of replacing an NPB allele with a 93-11 allele; D, dominance effect; R2, proportion of phenotypic variance explained by the QTL effect.

**Table 5 ijms-23-08296-t005:** Phenotypic performance of TGW in twelve NILs in BC_9_F_5_.

Line	Mean	SD	LSD Test
NPB	26.76	0.26	A
N83	28.07	0.49	B
R1	27.07	0.45	A
R2	26.77	0.47	A
R3	26.91	0.34	A
R4	26.7	0.63	A
R5	26.86	0.17	A
R6	26.95	0.28	A
R7	26.79	0.24	A
R8	28.09	0.33	B
R9	28.15	0.13	B
R10	27.86	0.38	B
R11	28.27	0.52	B
R12	27.99	0.39	B

**Table 6 ijms-23-08296-t006:** Phenotypic analysis of TGW of NPB and 93-11 homozygotes in five recombinant population in BC_9_F_5_.

Population	Haplotype	Mean	SD	Number of Plants	*p* Value
N1	NPB	28.12	0.36	15	0.4411
	93-11	28.20	0.44	19	
N2	NPB	26.80	0.31	17	<0.0001
	93-11	28.07	0.22	21	
N3	NPB	26.88	0.26	20	<0.0001
	93-11	28.15	0.31	18	
N4	NPB	28.09	0.45	23	0.4829
	93-11	28.17	0.30	17	
N5	NPB	28.23	0.46	19	0.9606
	93-11	28.11	0.45	19	

Student’s *t*-test was used to generate the *p* values.

**Table 7 ijms-23-08296-t007:** Candidate genes in the target region of qTGW7b.

Candidate Gene	Gene Product	Polymorphic Information in CDS	Amino Acid Variation
*LOC_Os07g39350*	Transporter family protein	None	None
*LOC_Os07g39360*	Transporter family protein	None	None
*LOC_Os07g39370*	Expressed protein	4-bp insertion, 17-bp deletion, 1 SNP	Code shift beginning at the 47th amino acid
*LOC_Os07g39380*	Retrotransposon protein	/	/
*LOC_Os07g39390*	Retrotransposon protein	/	/
*LOC_Os07g39400*	Polyol transporter 5	12-bp insertion, 4 SNPs	K123R, PAPA insertion
*LOC_Os07g39410*	Retrotransposon protein	/	/
*LOC_Os07g39420*	Retrotransposon protein	/	/
*LOC_Os07g39430*	mTERF family protein	5-bp deletion in the first intron	L119V, Q424L
*LOC_Os07g39440*	Zinc finger protein	4 SNPs	R210G, L356S, S367P
*LOC_Os07g39450*	Expressed protein	4 SNPs	C28R, R65W, K66E
*LOC_Os07g39460*	Transporter family protein	None	None
*LOC_Os07g39470*	chitin-inducible gibberellin-responsive gene, CIGR2	2 SNPs	None
*LOC_Os07g39480*	WRKY transcription factor, OsWRKY78	42-bp deletion, 4-bp deletion, 3 SNPs	P163L, V186I, M478I, N579S, amino acid deletion at 479-492 and 543

## Data Availability

The data presented in this study are available on request from the corresponding author.

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
