# Peer review of "Fine Mapping of qTGW7b, a Minor Effect QTL for Grain Weight in Rice (Oryza sativa L.)"

_ijms, 2022, doi:10.3390/ijms23158296_

Round 1

Reviewer 1 Report

The manuscript is well written and very detailed in the discovery of genes underlying the QTL effect. It needs minor changes. My only concern is a mistake in line 205 (instead of the G31 and G31 markers...the G31 and G32 markers). Otherwise, the manuscript is very sound.

Author Response

Sorry for this mistake, and we have corrected it. (line 205)

Reviewer 2 Report

Dear authors,

I really like the paper, although I would have some second toughts on the title saying minor effect QTL. There is an unknown number of minor QTL involved in formation of this trait, making only a single component of grain yield. However, after reading the results, I am convinced.

Minor comments:

L94: reference to previous study should be after mention

First paragraph of  Introduction section reads poorly and whole manuscript

Change „Number of values“ in all histograms to Distribution density

It would be nice to have a separate conclusions section drawing the main aims, cfindings and perspectives

Reviewer 3 Report

I am interested in the results of the study entitled Fine-Mapping of qTGW7b, a Minor Effect QTL for Grain 2 Weight in Rice (Oryza sativa L.).’ I inform you that it can be printed for the following reasons.

I find it very interesting to study the insignificant causes of rice grains as key traits for quality and yield through QTL analysis.

Author Response

We appreciate your acknowledgement of our efforts and wish you the best!
